

# The role of acceptance and values in quality of life in patients with an acquired brain injury: a questionnaire study

Gunther Van Bost[1,2], Stefaan Van Damme[1] and Geert Crombez[1]

[1] Department of Experimental-Clinical and Health Psychology, Ghent University, Ghent, Belgium
[2] Unit Acquired Brain Injury, Centrum voor Ambulante Revalidatie Ter Kouter, Deinze, Belgium

## ABSTRACT

**Objective**. An acquired brain injury (ABI) is a challenge for an individual's quality of life (QOL). In several chronic illnesses acceptance has been found to be associated with a better health-related quality of life. This study investigated whether this relationship is also found in patients with ABI. We also explored the impact of the perceived ability to live according to one's own values (life-values-match).

**Methods**. A total of 68 individuals (18–65 years of age) with an acquired brain injury completed a battery of questionnaires. The relations between health-related QOL (SF-36) and disease specific QOL (EBIQ; European Brain Injury Questionnaire), and personal values (Schwartz Values Inventory) and acceptance (ICQ; Illness Cognitions Questionnaire) were investigated. An additional question measured the life-values-match. Rehabilitation professionals reported the extent of impairment involved.

**Results**. Acceptance was positively associated with mental aspects of health-related QOL and the EBIQ Core Scale, after demographic variables and the extent of impairment were introduced in the regression. In a post hoc analysis we found that the life-values-match mediated the relationship between acceptance and mental aspects of QOL.

**Conclusion**. In patients with an ABI, promoting acceptance may be useful to protect QOL. Strengthening the life-values-match may be a way to accomplish this.

## INTRODUCTION

An acquired brain injury (ABI) may well be one of the greatest challenges to live with (*Seibert et al., 2002*). A systematic review (*Polinder et al., 2015*) reported a high prevalence of health problems during the first year after the injury, and even in the long-term patients show large deficits in mobility, communication and cognitive functions. People may suffer from motor disorders such as hemiplegia, but also experience aphasia or attention and memory problems.

Interestingly, health-related quality of life (HRQOL), here defined as the perception of how illness and treatment affect physical, mental and social aspects of life (*Dijkers, 2004*) is only weakly related to the severity of impairment in ABI (*Grauwenmeijer, Heijenbrok-Kal*

Corresponding author
Gunther Van Bost,
gunther.vanbost@ugent.be

& Ribbers, 2014). This observation may indicate that other variables affect HRQOL. One such variable that may account for HRQOL despite adversities is the way patients cope with their problem.

Coping is an elusive construct, and there are myriad ways of measuring and classifying coping strategies. After an extensive review of the literature, Skinner et al. (2003) concluded that the dual-model of coping (Brandtstädter & Rothermund, 2002) was exemplary for its scope and clarity. The dual-process model was originally developed to describe the self-regulatory processes in an ageing population, but has been widely adopted to cope with various adversities such as chronic pain (Lauwerier et al., 2010) or chronic fatigue syndrome (Van Damme et al., 2006). According to this model, the first mode of coping with adversity is to identify the factors that hinder goal pursuit and to attempt to reduce or eliminate the obstacle. This is called 'assimilative' coping. When an obstacle is soluble, this type of problem solving is considered adaptive. However, when obstacles are uncontrollable, a slavish adherence to assimilative coping may prove futile and only increase frustration and distress. 'Accommodative' coping may then be preferred. During this type of coping patients accept that the problem cannot be resolved, disengage from the pursuit of the blocked goal, and finally engage in new or reset goals according to one's values (Wolters-Grégorio et al., 2010).

Without doubt, assimilative coping can be a useful strategy in ABI. Rehabilitation has proven effective, and studies indicate that more intensive rehabilitation is associated with more rapid gain (Turner-Stokes, 2008). Nevertheless, a complete return to the pre-injury status is often unlikely. Individuals with severe memory problems will not be able to follow higher education; and a teacher with global aphasia will experience difficulties to teach. Although problem solving strategies are considered adaptive (Anson & Ponsford, 2006; Wolters-Grégorio et al., 2010), an accepting attitude towards problems that cannot be resolved may also prove to be beneficial. It should be clear that acceptance is not resigning or giving up (Risdon et al., 2003). It is the acknowledgement that a problem is not likely to disappear, and it is better to shift the attention and efforts towards other aspects of life according one's personal values. According to Klonoff (2010), acceptance after brain injury means "patients' ability and willingness to cope with their new reality and identity" (p. 100).

The role of acceptance has been documented in various chronic illnesses amongst which chronic pain (Viane et al., 2004), chronic fatigue syndrome (Van Damme et al., 2006), multiple sclerosis (Pakenham, 2006), and chronic kidney disease (Poppe et al., 2013). Evidence has accumulated that attempts to control uncontrollable events may bring along cognitive and emotional costs, whereas accepting the uncontrollable nature of illnesses allows for a re-engagement in feasible activities (Wrosch et al., 2003). Research on acceptance in patients with ABI is limited, but promising. One study (Snead & Davis, 2002) has revealed a positive relationship between acceptance, measured with the Acceptance of Disability Scale (Linkowski, 1971), and HRQOL in individuals with an ABI ten years post-injury. Also, acceptance has been associated with less depression, after controlling for demographic factors in patients with stroke (Townsend et al., 2010). These results require replication and corroboration.

Another question pertains to how exactly acceptance may lead to a better HRQOL. Human behavior is often guided by values, which have been found to be largely consistent across cultures (*Schwartz & Boehnke, 2004*). On the one hand, certain values may directly lead to behavior or cognitions that enhance one's HRQOL. For example, students having certain values (i.e., achievement, stimulation and self-direction) may report a more positive sense of wellbeing than when they have other values (e.g., conformity and security) (*Sagiv & Schwartz, 2000*). On the other hand, being successful at living according to personal values, regardless of what those values are, may be essential for HRQOL. For example, chronic pain patients who report to live according to their values reported less disability, depression, and pain-related anxiety (*McCracken & Yang, 2006*). It may be that individuals experience the impairments by the brain injury as less distressing as long as these do not interfere with their life values.

## Study objective

This study was designed to investigate the role of acceptance in the HRQOL in patients with ABI. First, we aimed at replicating and extending the finding that self-reported acceptance is associated with higher scores on measures of HRQOL, using the Short Form Health Survey (SF-36) (*Ware & Gandek, 1998*) and the European Brain Injury Questionnaire (EBIQ) (Supplemental Information 1; *Teasdale et al., 1997*). We expect to see similar results as previous studies with chronic medical conditions: acceptance of illness is related to a higher HRQOL. Nevertheless, we expect a stronger effect on the mental component than on the physical component of HRQOL, as we consider it unlikely that acceptance will alter for example the self-reported ability to walk again or climb the stairs. Second, we investigate the specific role of values. Given the fact that in a healthy population certain values (i.e., achievement, stimulation and self-direction) were associated with more positive outcomes, we expected a similar pattern in ABI. Also, individuals who consider themselves as part of a larger reality (e.g., nature, humanity) will have a high score on the value Universalism, and may find it easier to focus on a positive project, even with their disability. Third, we also explored whether living according one's own values (i.e., life-values-match), independent from the specific value, was associated with more positive outcomes.

## MATERIALS AND METHOD

### Participants

Sixty-eight persons with an acquired brain injury participated in this study. A large majority ($N = 58$) was recruited from three outpatient rehabilitation units in Flanders (Dutch-speaking region in the north of Belgium); four patients from a specialized psychiatric unit; and four from a private practice of a specialised psychotherapist. These outpatient rehabilitation units work with ABI-patients, regardless of the aetiology of the brain injury, to maximize their level of activities and participation as defined by the International Classification of Functioning, Disability and Health of the World Health Organisation (*Bilbao et al., 2003*). The study protocol was approved by the Ethical Committee of the Faculty of Psychology and Educational Sciences of Ghent University

(2006/39). All patients provided written informed consent. When there was doubt about the ability of a patient to make autonomous decisions about the participation, a relative was asked to provide additional consent; this happened twice. A graduate student was present during the study to help the respondents to stay focused, to provide explanation when the respondents did not understand the question and to provide practical help to fill out the questionnaires when necessary.

### Study requirements

This study required that patients had at least a basic level of awareness about the consequences of the brain injury, although it was not necessary that they could provide a precise description of these impairments. During the interview, it became clear that two participants did not meet that criterion because they were guessing or providing examples that were unrelated to the question. Therefore, they were excluded from further analysis. Therefore, the final sample consisted of 68 patients (41 male and 27 female), mean age = 46.3 years (SD = 14.6; range: 18–68). Mean time since brain injury was 25.8 months (SD = 27.9; range 3–144). None of the participants was employed, although a few did volunteer work. 35.7 % had lower or vocational education, 28.6 % had middle education, 21.4 % received higher non-university education, and 12.5 % had university education. Except for the four respondents in a specialised psychiatric unit, there was sufficient support from the family to allow them to remain in their home environment. A total of 33 respondents had a stroke, 30 had a traumatic brain injury and five respondents had a brain injury following anoxia after heart failure.

### Questionnaires

Acceptance was measured using the acceptance subscale of the Dutch Illness Cognition Questionnaire (ICQ) (*Evers et al., 2001*). The ICQ is an 18-item self-report instrument assessing: (1) Helplessness (6 items, e.g., "My illness frequently makes me feel helpless"), (2) Acceptance (6 items, e.g., "I have learned to live with my illness") and (3) Disease benefits (6 items, e.g., "My illness has made me appreciate life more"). Items are rated using a 4-point scale (1 = "not", 2 = "a little", 3 = "strongly", 4 = "completely"). The ICQ hasn't been used before in a brain-injured population, but the three-factor structure and psychometric properties have been found to be good in a Dutch-speaking population of persons with chronic pain and chronic fatigue (*Lauwerier et al., 2010*).

The Schwartz Values Questionnaire—Dutch version (SVQ) (*Schwartz & Boehnke, 2004*) measured specific personal values. It consists of 58 items and 10 values: (1) Power (5 items, e.g., "social power"), (2) Achievement (5 items, e.g., "successful"), (3) Hedonism (3 items, e.g., "enjoying life"), (4) Stimulation (3 items, e.g., "an exciting life"), (5) Self direction (5 items, e.g., "choosing own goals"), (6) Universalism (8 items, e.g., "equality"), (7) Benevolence (5 items, e.g., "helpful"), (8) Tradition (5 items, e.g., "respect for tradition"), (9) Conformity (4 items, e.g., "politeness"), and (10) Security (5 items, e.g., "family security"). Each item is followed by a short explanatory phrase (e.g., FAMILY SECURITY (security for those you love)). Patients are asked to rate the importance of each value item as guiding principle in their life on a 9-point scale, ranging from −1 ("opposed to my

principles"), through 0 ("not important"), to 7 ("of supreme importance"). In a large international study (*Schwartz & Boehnke, 2004*) the 10-factor structure has been confirmed and the psychometric properties were satisfactory. As far as we know, this questionnaire hasn't been used before with brain injured patients.

At the end of the SVQ, respondents were asked to what extent they felt able to live according to their own personal values. This "life-values-match" was specifically developed for this study. Participants responded on a single 7-point scale (1 = no match at all; 7 = perfect match between the actual life and valued life) to what extent they were overall able to live according to values, mentioned in the SVQ". This "life-values-match" should be seen as an extension of the SVQ rather than as an independent instrument. Respondents have just answered 58 questions concerning values and were then asked about their ability to live according to those values. Without these previous questions, this item may lose its meaning or understandability.

Quality of life was measured by the Dutch version of the Short Form Health Survey (SF-36) (*Ware & Gandek, 1998*) and by the Dutch version (Supplemental Information 1) of the European Brain Injury Questionnaire (EBIQ) (*Teasdale et al., 1997*). The SF-36 consists of 36 items, and is recommend by *Polinder et al. (2015)* as a generic measure of QOL in patients with TBI. This study reports an internal consistency ranging from fair to good and a good content validity across various studies. The SF-36 yields an 8-scales health profile, and two components scores: a physical health component (e.g., Accomplished less as a result of your physical health) and a mental health component (e.g., Did work or activities less carefully than usual as a result of emotional problems.). *Bullinger & The TBI Consensus Group (2002)* recommended the EBIQ as a disease-specific instrument for QOL-research with a brain-injured population. *Teasdale et al. (1997)* derived 8 scales: (1) Somatic (8 items, e.g., "Lack of energy"), (2) Cognitive (13 items, e.g., "Trouble concentrating"), (3) Motivation (5 items, e.g., "Lack of interest in hobbies in the home"), (4) Impulsivity (13 items, e.g., "Behaving tactlessly"), (5) Depression (9 items, e.g., "Feeling hopeless about the future"), (6) Isolation (4 items, e.g., "Thinking only of oneself"), (7) Physical (6 items, e.g., "Needing help with personal hygiene", (8) Communication (4 items, e.g., "Difficulty in communication") and (9) Core (34 items, e.g., "Problems in general"). To obtain a single indicator of disease specific QOL, we used the Core Symptoms scale, which consisted of the most sensitive items from the 8 subscales (e.g., Lack of energy or being slowed down) to be rated on a 3-point scale ("not at all", "a little", "a lot"). The first and the last author, and two other Dutch speaking persons, translated the EBIQ in Dutch in 2005. The first author made a back-translation and asked Prof. Teasdale to check the back-translation. He had some minor remarks that have been addressed in the final version. Reliability and validity of the English version of the SF-36 (*Findler et al., 2001*) and the EBIQ (*Sopena et al., 2007*) have proven satisfactory in a sample of brain-injured patients. This is the first time that the Dutch version of the questionnaire is used.

The therapist who was responsible for the rehabilitation program, provided four expert ratings, respectively for the level of motor impairment, communication impairment, cognitive impairment and self-awareness impairment. For each impairment, a 7-point scale (7 = perfect age-appropriate functioning, 1 = extremely impaired) was used.

## RESULTS

Data were checked for normality and we didn't find violations of the assumptions for further analyses. The sample was a convenience sample, and no power calculation was used. Mean scores, standard deviations, internal consistency (Cronbach's α) for acceptance, the life-values-match and the different indicators of HRQOL are presented in Table 1. A significant difference between the results of participants with a TBI and those after a stroke was observed for the Physical Component of the SF-36 ($t(61) = -2.06$; $p < .05$). There were no significant differences found for the Mental Component ($t(61) = -1.17$; ns) or the EBIQ Core ($t(61) = -.58$; ns). As statistical power was low we did not include aetiology in further analyses.

Pearson correlations can be seen in Table 1. None of the demographic factors had a significant correlation with Mental Component of the SF-36 or disease specific HRQOL. Male gender reported more acceptance (ICQ Acceptance, $t(66) = 2.11$; $p < .05$) and a higher physically quality of life (Physical Component of the SF-36, $t(66) = 2.26$; $p < .05$). The difference between men and women for the ICQ Acceptance scale is 2.38 points (). For the Physical Component the difference was 5.39 ($t(66) = 2.26$; $p < .05$). Age was only negatively correlated with the Physical Component of the SF-36. Education was related with the Physical Component ($F(10, 56) = 2.06$; $p < .05$), but not with the other indicators of QOL of Acceptance. Self-awareness, as rated by the therapist, was negatively correlated with the Physical Component of the SF-36 and positively with Acceptance. Motor problems correlated negatively with the Physical Component of the SF-36. Communication problems were negatively correlated with the Physical Component of the SF-36 and also negatively correlated with Acceptance. Cognitive problems had a negative correlation with the life-values-match. Acceptance was positively related to the Physical and the Mental Component of the SF-36 and was negatively related to the EBIQ Core Scale. The only scale of the Schwartz Values Inventory that was related to acceptance was Universalism. None of the scales of the Schwartz Value Inventory were related to HRQOL measures. However, the single item life-values-match was strongly related to the SF-36, disease-specific HRQOL and acceptance.

The role of Acceptance in HRQOL was investigated by a series of multiple regression analyses, with the Physical and Mental component of the SF-36 and the EBIQ Core Scale as dependent variables. In each analysis age, gender and education were entered in a first step. In a second step, the time since injury was entered. The four expert ratings of the illness characteristics were entered in the third step. In the fourth and final step, acceptance was entered. The results of the final model of these analyses are shown in Table 2. In the analysis with the Physical Component (SF-36) as the dependent variable the outcome was significantly higher for male gender ($\beta = 0.21$, $p < 0.05$), and with less severe Motor impairments ($\beta = -0.54$, $p < 0.001$), and a lower Self-Awareness ($\beta = -0.25$, $p < 0.05$). The impact of Acceptance approached significance ($F_{\text{change}}(1, 57) = 3.73$, $p < .058$). $R^2$ change after introduction of Acceptance was 0.03. The final model explained 45% of the variance in the SF-36 Physical Component scores.

Peer J

**Table 1  Correlations between indicators of HRQOL, Acceptance, Life-Values-Match, illness characteristics and demographics.**

| Scale | Mean (SD) | Cronbach's α | 1 | 2 | 3 | 4 | 5 | 6 | 7 | 8 | 9 | 10 | 11 | 12 |
|---|---|---|---|---|---|---|---|---|---|---|---|---|---|---|
| 1. SF-36 Physical | 41.8 (9.9) | .89 | – | – | – | – | – | – | – | – | – | – | – | – |
| 2. SF-36 Mental | 61.8 (12.1) | .81 | .44** | – | – | – | – | – | – | – | – | – | – | – |
| 3. EBIQ Core | 55.1 (12.8) | .92 | −.35* | −66*** | – | – | – | – | – | – | – | – | – | – |
| 4. Acceptance (ICC) | 14.7 (4.7) | .85 | .25* | .43*** | −.47*** | – | – | – | – | – | – | – | – | – |
| 5. Life-values -match | 4.4 (1.7) | – | .31* | .47*** | .41*** | .52*** | – | – | – | – | – | – | – | – |
| 6. Self-awareness | 5.6 (1.6) | – | −.36** | −.07 | .01 | .31* | .22 | – | – | – | – | – | – | – |
| 7. Motor problems | 3.8 (1.7) | – | −.58*** | −.03 | −.04 | −.19 | −.17 | .25 | – | – | – | – | – | – |
| 8. Cognitive problems | 3.4 (1.3) | – | .11 | −.08 | .18 | −.15 | −.15 | −.32** | −.09 | – | – | – | – | – |
| 9. Communication problems | 2.7 (1.8) | – | −.25* | −.06 | .10 | −.24* | −.10 | .05 | .42*** | −.07 | – | – | – | – |
| 10. Gender (male) | – | – | .27* | .11 | −.06 | .25* | .02 | .04 | −.01 | −.02 | −.21 | – | – | – |
| 11. Age (years) | 46.1 (14.7) | – | −.30* | −.07 | −.05 | .09 | .13 | .35 | .13 | .16 | .10 | .03 | – | – |
| 12. Education (years) | 12.4 (2.9) | – | .03 | .16 | −.04 | −.01 | −.09 | −.03 | .13 | .08 | −.09 | .18 | −.06 | – |
| 13. Time since injury (months) | 25.6 (27.8) | – | .14 | −.02 | .19 | −.02 | .04 | −.18 | −.23 | −.15 | −.12 | −.01 | −.25* | −.19 |

**Notes.**

*p < .05.
**p < .01.
***p < .001.

**Table 2  Hierarchical regression analyses on different indicators of HRQOL (final model).**

| Dependent variable | Step | Predictors | β (standardized) | $\Delta R^2$ | $R^2$ (adjusted) |
|---|---|---|---|---|---|
| SF 36 Physical | 1 | Gender | .21[*] | .15[*] | .11 |
| | | Age | −.18 | | |
| | | Education | .11 | | |
| | 2 | Time since injury | −.03 | .01 | .10 |
| | 3 | Self-awareness | −.24[*] | .34[**] | .43 |
| | | Motor problems | −.54[**] | | |
| | | Cognitive problems | −.04 | | |
| | | Communication problems | .10 | | |
| | 4 | Acceptance | .20 | .03 | .45 |
| SF 36 Mental | 1 | Gender | −.01 | .04 | −.01 |
| | | Age | −.04 | | |
| | | Education | .17 | | |
| | 2 | Time since injury | .04 | .00 | −.02 |
| | 3 | Self-awareness | −.26 | .02 | −.07 |
| | | Motor problems | .07 | | |
| | | Cognitive problems | −11 | | |
| | | Communication problems | .08 | | |
| | 4 | Acceptance | .53[**] | .22[**] | .16 |
| EBIQ Core | 1 | Gender | .07 | .01 | −.04 |
| | | Age | −.04 | | |
| | | Education | .03 | | |
| | 2 | Time since injury | .18 | .04 | −.02 |
| | 3 | Self-awareness | .33[*] | .04 | −.03 |
| | | Motor problems | −.20 | | |
| | | Cognitive problems | .15 | | |
| | | Communication problems | .07 | | |
| | 4 | Acceptance | −.57[**] | .25[**] | .23 |

Notes.
[*] $p < .05$.
[**] $p < .001$.

In the analysis with the Mental Component (SF-36) as dependent variable none of the demographic factors or illness characteristics produced a significant effect. The Mental Component (SF-36) was only positively accounted for by Acceptance ($F_{change}(1, 57) = 16.95$, $p < .001$). $R^2$ change after introduction of Acceptance was 0.22. The final model explained 16 % of the variance in the SF-36 Mental Component scores.

Also in the analysis with the EBIQ Core Scale as dependent variable there was no effect of demographic variables, but there was a positive effect of Self-awareness ($\beta = 0.33$, $p < 0.05$). Acceptance had an unique explanatory value ($F_{change}(1, 57) = 21.01$, $p < .001$) beyond the other variables. A higher acceptance was linked with less disease specific complaints. $R^2$ change after introduction of Acceptance was 0.25. The final model explained 23 % of the variance in the EBIQ Core Scale scores.

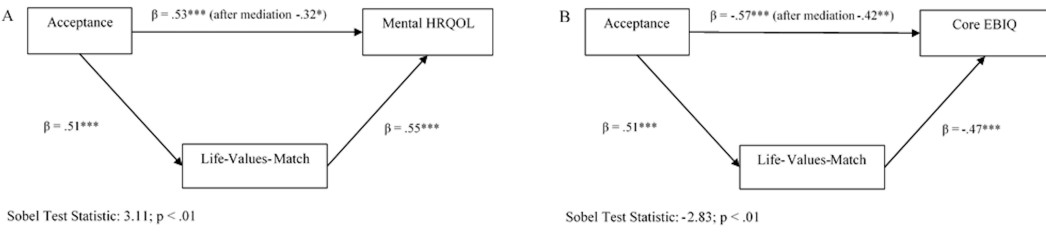

**Figure 1  Mediation of life-values-match between acceptance and QOL.**

In further post-hoc exploratory analyses, we investigated whether the life-values-match was a mediator of the relationship between acceptance and HRQOL. A mediator is "a variable, that serves to explain the process by which a predictor significantly affects an outcome, such that the predictor is associated with the mediator, which in turn is associated with the outcome" (*Holmbeck, 2002*). To test for mediation, the following conditions should be met: (a) a significant association between the predictor and the outcome, (b) a significant association between the predictor and the mediator, and (c) a significant association between the mediator and the outcome, after controlling for the effect of the predictor. If these conditions are met, then one examines whether the predictor-outcome-effect is less after controlling for the mediator. The Sobel-test, which is basically a specialized $t$-test, is used to determine if this reduction in effect is significant. These conditions were only met for the effect of acceptance on the mental component of the SF-36 and on the Core Scale of the EBIQ. Hence, a mediation analysis was only performed for these associations. As shown in Fig. 1 we found that the life-values-match significantly mediated the relationship between Acceptance and the Mental Component of HRQOL (SF-36). The remaining predictive value of Acceptance in explaining the Mental Component of HRQOL was significantly reduced by the inclusion of the life-values-match (Sobel Test Statistic = 3.11, $p < .01$). Also the mediation effect of the life-values-match on the relationship between Acceptance and the Core Scale of the EBIQ was significant (Sobel Test Statistic = $-2.83$, $p < .01$).

## DISCUSSION

This study revealed that acceptance was uniquely associated with measurements of general and disease specific quality of life in ABI patients. This is in line with the work of *Snead & Davis (2002)*, who concluded that greater acceptance of disability was associated with higher quality of life in a sample of 40 individuals with an acquired brain injury. At first sight this may be at odds with the results of *Wolters-Grégorio et al. (2010)*, who found that an active problem-focused coping style is associated with a higher quality of life in a sample of 110 individuals in the chronic phase after brain injury, whereas more passive emotion-focused coping styles turned out to be more maladaptive. However, acceptance is not to be understood as a passive, emotion-focused process, but a way of coping in which individuals disengage from unattainable goals and pursue more feasible goals.

Striving for personal goals assigns meaning, structure and direction to an individual's life and is known to be associated with wellbeing (*Conrad et al., 2010*). According to

*Brandtstädter & Rothermund (2002)* when people get older, they invest less in trying to solve the problems that block their goals, and invest more in the adaption of their goals so that these become achievable. In a similar way, a brain injury blocks individuals' personal goals, causing distress. One way of coping with such distress is to attempt restoring the status and functioning as before the injury. Patients may then engage in intensive rehabilitation efforts, retraining the damaged functions to be able to overcome activity and participation restrictions (*Bilbao et al., 2003*). At a certain point this strategy is no longer useful, because certain impairments are impossible to overcome. Keep fighting these impairments may then lead to more frustration and a lower HRQOL. If acceptance is considered part of accommodative coping, one may understand how this type of coping contributes to a better mental wellbeing. *Conrad et al. (2010)* have found similar results about the impact of the attainability of life goals on subjective wellbeing in a brain-injured population.

A particular challenge in patients with ABI is that some brain injury related problems, such as cognitive inflexibility and low self-awareness may complicate a shift to accommodative coping. Especially the lack of cognitive flexibility may lead to perseveration and difficulties disengaging from unattainable goals. However, *Wolters-Grégorio et al. (2015)* found no relationship between measures of coping and life satisfaction and neuropsychological test results of executive functioning, although there was a relationship with self-reported problems in executive functioning. Further research is needed to clarify this relationship.

This study demonstrated the importance of values. Of particular interest is the finding that the relationship between acceptance and mental HRQOL was mediated by the perceived ability to live according to one's values. Possibly, important changes in life because of a brain injury are easier to accept as long as the new life is still in concordance with one's values. For example, a former engineer who strongly values professional success and being respected by others may experience a good quality of life by growing and selling vegetables on a small scale, as long as he feels successful and respected doing so.

We found no evidence that certain specific values were superior to others in explaining quality of life, although there was a correlation between Acceptance and Universalism. This is in line with the findings of *Sagiv & Schwartz (2000)* in a student population, who also did not find that particular values had an effect upon wellbeing. Of further note is that the size of the association between acceptance and the physical HRQOL is much smalle than the one between acceptance and the mental component. This finding was not unexpected. The self-perceived physical capabilities of brain-injured individuals are probably largely determined by demographic factors and the impairments of the brain injury (*Berzina et al., 2013*), probably leaving not much room for effects of coping.

A better understanding of the determinants of the acceptance process can contribute to the development of intervention techniques, aimed at a better quality of life of ABI patients. More research is needed to study these processes. Acceptance and Commitment Therapy (ACT) (*Hayes et al., 2006*) has made acceptance an important focus of therapeutic interventions. In chronic pain patients, there is evidence that therapeutic interventions aimed at acceptance and values-based-action (*Vowles & McCracken, 2008*) are effective. We may expect similar results with a brain-injured population, given that in post hoc

analyses we found that the relationship between acceptance and mental QOL and between acceptance and the disease-specific QOL was mediated by the perceived ability to live according to one's values. This is also found in the few available studies that have investigated the possibilities of ACT with an ABI-population (*Kangas & McDonald, 2011*; *Whiting et al., 2012*).

This study has some limitations. First, a cross-sectional design was used, which makes causal inferences impossible. The reversed direction is also possible: people experiencing an overall higher HRQOL may be more able at withstanding adversity and may find it easier than others to accept this reality. Second, impairment was only assessed by the therapist and only in four areas. Although patients with a severely impaired self-awareness were excluded from our study, several respondents were less able to report the consequences of their brain injury, minimized them or could not understand their impact. A correct appreciation of the situation might be necessary for the acceptance process, resulting in a positive relationship between self-awareness, life-values match and acceptance. Third, we only investigated HRQOL, ignoring life satisfaction or other aspects of QOL after brain injury (*Dijkers, 2004*). The impact of psychological factors as acceptance on life satisfaction may be stronger, knowing that the role of demographic factors and impairments is very limited there (*Pierce & Hanks, 2006*). Fourth, we need to be cautious with the interpretation of the results with the Schwartz Values Inventory (SVI). We observed that for many patients the abstract phrasing in the questionnaire was difficult, even with help. We experienced that many patients with ABI had to be reminded of the distinction between the values as a guiding principle in their life (e.g., being active) and the actual status of being active. We tried to compensate for this with the help of a graduate student as a research assistant. The graduate student was not blind to the research objectives and we may not exclude the possibility that this affected the results. Fifth, the sample size was moderate, making it difficult to perform subgroup analysis. We chose to include demographic factors and illness characteristics in the analysis anyway, as it is still a common idea that these factors are important for the mental aspects of subjective quality of life. By showing that such a relation is hard to find, we hope that one will put more emphasis on other factors, such as acceptance.

Despite these limitations this study has revealed similar effects of acceptance as were previously observed in other chronic conditions. It also suggests the importance of reducing the discrepancy between the valued way of living and the actual way of living in protecting patients' HRQOL. These findings are useful for the development of clinical interventions, specifically aimed at an ABI-population. When complete recovery is no longer feasible, it may be useful to assess the basic values of patients. This can help therapists to guide people in their search for other meaningful activities in life.

### Funding

The authors received no funding for this work.

### Competing Interests

The authors declare there are no competing interests.

### Author Contributions

- Gunther Van Bost conceived and designed the experiments, performed the experiments, analyzed the data, contributed reagents/materials/analysis tools, wrote the paper, prepared figures and/or tables, reviewed drafts of the paper.
- Stefaan Van Damme and Geert Crombez conceived and designed the experiments, analyzed the data, contributed reagents/materials/analysis tools, wrote the paper, reviewed drafts of the paper.

### Human Ethics

The following information was supplied relating to ethical approvals (i.e., approving body and any reference numbers):

The Ethical Committee of the Faculty of Psychology and Educational Sciences of Ghent University gave written approval under ref. 2006/39.

### Data Availability

The raw data has been supplied as Data S1.

### Supplemental Information

Supplemental information for this article can be found online at http://dx.doi.org/10.7717/peerj.3545#supplemental-information.

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
