# Peer review of "The role of acceptance and values in quality of life in patients with an acquired brain injury: a questionnaire study"

_PeerJ, doi:10.7717/peerj.3545_

## Round 0.1 · original submission · Minor Revisions

Dear Authors,

There are quite a number of excellent suggestions to improve your manuscript from the three peer reviewers that have read your manuscript. Please revise and re-submit the manuscript for re-review.

Reviewer 1 ·

Basic reporting

Overall 'basic reporting' is adequate. Suggestions for improvement (see especially the one regarding line 169):
* Abstract: Is there a word missing in the last sentence regarding the Methods, after "the rehabilitation professionals” (e.g. ‘involved.’)?
* Line 5: “large deficits” can you give a few examples?
* Line 58: the use of the word “by” is unclear; can it be deleted?
* Line 66: “philosophical” values have not been introduced as a concept; not everyone will immediately know what universalism is in this context. It would be helpful to expand the introduction in this regard.
* Line 85-88: Do you mean that two people were excluded because they did not have a minimal level of self-awareness regarding the consequences of their brain injury? It is not entirely clear to me what “Nevertheless” refers to.
* Line 137-138: i.e. this measurement involved 4 items?
* Line 151: “the most important” why/in what respect was it the most important?
* Line 157: What does “these” refer to?
* Line 165: Should it be “variable” or “variables”?
* Line 169: “the outcome was significantly higher” >> it is surprising that higher quality of life is connected to more motor & communication impairment, and it also seems to be partially at odds with the information in your tables: quality of life seems to be related to fewer rather than increased motor problems. Please double-check all descriptions and adjust as needed.
* Line 175: “and” should probably be “or”
* Line 265: “abler” I’m not sure this is correct English, I’d use “more able”
* Table 1: first variable says "111" instead of "1"

Experimental design

* Line 81: Was the graduate student aware of the hypotheses? If yes, how likely is it that he/she has influenced the responses?
* Line 186: it seems that these are post-hoc analyses, is that correct? If so, it would be good to mention this explicitly at this location, in the abstract, and in the discussion.

Validity of the findings

Please see the comments above. In addition, please ensure that the datafile is entirely in English and that the variable labels are added (in a separate worksheet or file).

Additional comments

Interesting to look at acceptance and values! All the best for the continuation of your research.

Reviewer 2 ·

Basic reporting

This is an interesting and well-researched paper. Paper was written clearly with the use of proper and professional English. Worth for publication as it provide new findings in the research area.

The use of tables and figures are in line with the paper content and discussion.

Experimental design

Not related as this piece of work is very much a survey work using various questionnaire instruments.

Validity of the findings

Suggest author to indicate in more detail on the use of each instruments. Are all of the standardized instruments (scales) used in this research receive approval from the original developer?

Author have indicated the development of the"life-values-match" instrument in the paper. What is the reliability and validity score for the instrument?

Suggest to elaborate further on this two aspects as it will ensure the validity of the study.

Additional comments

Suggest author to include sub-title "Study Objective" - line 55.

Suggest author to include sub-title "Study Criteria" - line 85.

Suggest author to expand a little more on the study implication and possible clinical implications at the end of the discussion section.

Reviewer 3 ·

Basic reporting

The introduction and writing overall had a very nice flow, clearly showing the rationale for the study. The writing was engaging for the reader.

The writing was generally clear, although some minor grammatical proofreading and modifications are recommended, such as use of was/were, etc. Some examples are given below.

• A systematic review [25] reported a high prevalence of health problems during the first year after the injury, and even in the long-term patients show large deficits. – Please clarify the term "large deficits".
• One such variable that may account for a HRQOL despite adversities is the way patients cope with their problem. – Consider to remove "a" before HRQOL
• One study [33] has revealed that a positive relationship between acceptance, measured with the Acceptance of Disability Scale [21], and HRQOL in individuals with an ABI ten years post-injury. – remove "that" after revealed so should be “has revealed a positive relationship”
• Another question pertains to how exactly acceptance may lead to a better HRQOL.—Have this start a new paragraph rather than having the paragraph be so long.
• example, students fostering certain values – consider to change the word "fostering" to another word, please change both times the word is used.

• 1. Typo on table for 1. it is listed in error as 111.

The references were sufficient except some more details could be added about the psychometric properties of the scales for past use and use in the Dutch language/context.

Experimental design

Overall a good experimental design. Some minor clarifications desired for the design but positively research was clearly within the aims and scope of journal and the research questions were well-defined.

Participants—it is unclear for an international audience where Flanders is located so please clarify the country region, etc. at first mention.

• All patients provided informed consent. – Was this written?

• When there was doubt about the ability of a patient to make autonomous decisions about the participation, a relative was asked to provide additional consent. – how frequently did this occur?
• A graduate student was present during the study to help the respondents to stay focused, to provide explanation when the respondents did not understand the question and to provide practical help to fill out the questionnaires when necessary. – how frequently was help needed and what level of help was needed, i.e., how frequently did participants not understand the question?

• This study required that patient had minimal level of self-awareness about the consequences of a brain injury, please change to

This study required that patient had at least a minimal level of self-awareness about the consequences of a brain injury,
Or
This study required that patient had at least a basic level of self-awareness about the consequences of a brain injury,
• None of the participants WERE employed, although a few did volunteer work.
• 35.7 % had lower or professional education, 28.6 % had middle education, 21.4 % received higher non-university education, and 12.5 % had university education. – These terms seem somewhat unclear for an international audience, by professional is it referring to vocational education?
Questionnaire
• Reliability and validity of the SF-36 [12] and the EBIQ [34] have proven satisfactory in a sample of brain-injured patients. – Please add a little more detail, and add about use of the scales in the Dutch context too.

Validity of the findings

Overall findings appear valid, though some minor improvements are recommended for improved clarity.
• Good that this information was analyzed that --there was no significant difference between the results of participants with a TBI and those after a stroke. Therefore, aetiology was not included in further analyses. – However best to state the significance level of such tests to determine the results were not significantly different, p < .05? and please state the type of analysis.
o Correlations – type of test, Pearson?
o So gender and education were analyzed as correlations, may also want to consider tests of group difference as well. Correlations can stay but group difference tests may also possibly be illuminating.
• Sixth, the sample size was moderate, making it difficult to perform subgroup analysis.
Please consider elaborating in a short paragraph rather than just a sentence on this point.
Limitations should also address the use of so many variables in the regression with an under-powered sample size. It is advised to consider entering only those variables with significant correlations into regression and/or alternatively acknowledging the limitations of so many predictors and explain the exploratory nature of the regression like what was done for the mediation.
* Consider reporting if power analysis was used.
* Consider more clear reporting of effect sizes, where relevant.
• It is suggested to very briefly mention initial assumption checking of the data, e.g., normality, and outliers if any.
• Consider reviewing the sentence in the abstract -- Acceptance was positively associated with mental aspects of HRQOL, after controlling for demographic variables and the extent of impairment.
The demographic variables were not significant so the term "controlled for” may lead the reader to have a different understanding of the data. The sentence can stay, but I recommend to consider revising if a better phrasing is decided upon, or if the regression is modified.

* It was excellent that possible clinical implications and applications of the study were discussed.

* Overall an excellent effort with minor revisions recommended.

Additional comments

It was a pleasure to read your paper. It was a very interesting study and I encourage follow-up work by yourselves in the area. Some minor writing improvement and clarifications of the methods/analysis is recommended. Your inclusion of the clinical implications of your study is noteworthy.

---

## Round 0.2 · accepted · Accept

Dear Authors,

Thank you for the revisions, that have led to the acceptance of the manuscript.

Congratulations!

Reviewer 1 ·

Basic reporting

Dear authors, thanks for the revisions that you have made. I have no new comments.

Experimental design

Idem

Validity of the findings

Idem

Additional comments

Dear authors, thanks for the revisions that you have made. I have no additional comments.

Reviewer 3 ·

Basic reporting

The revised is improved. There just remain occasional minor grammatical corrections that can be reviewed in the proof stage.

Experimental design

The design is clearer and much improved.

Validity of the findings

Again the findings are improved.

However, please do review the added new material about "gender" differences throughout the manuscript to double check all information was included and check the language flow. For example on line 190 it seems it is missing some text in the parentheses ( ).

Additional comments

Overall an excellent and interesting article. The revisions have now improved the manuscript.